# Characterization of the complete mitochondrial genome of *Desmaulus extinctorium* (Littorinimorpha, Calyptraeoidea, Calyptraeidae) and molecular phylogeny of Littorinimorpha

**Yanwen Ma**[1], **Biqi Zheng**[2], **Jiji Li**[1], **Wei Meng**[3], **Kaida Xu**[3]*, **Yingying Ye**[1]*

1 National Engineering Research Center for Marine Aquaculture, Zhejiang Ocean University, Zhoushan, 316022, China, 2 Department of Natural Resources, Ningde Marine Center, Ningde, 352000, China, 3 Key Laboratory of Sustainable Utilization of Technology Research for Fisheries Resources of Zhejiang Province, Zhejiang Marine Fisheries Research Institute, Scientific Observing and Experimental Station of Fishery Resources for Key Fishing Grounds, Ministry of Agriculture and Rural Affairs of China, Zhoushan, 316021, China

* h01011@zjou.edu.cn (KX); yeyy@zjou.edu.cn (YY)

**Data Availability Statement:** The datasets analysed during the current study are available in

## Abstract

For the purpose of determining the placement of Calyptraeidae within the Littorinimorpha, we hereby furnish a thorough analysis of the mitochondrial genome (mitogenome) sequence of *Desmaulus extinctorium*. This mitogenome spans 16,605 base pairs and encompasses the entire set of 37 genes, including 13 PCGs, 22 tRNAs and two rRNAs, with an evident AT bias. Notably, *tRNA*$^{Ser1}$ and *tRNA*$^{Ser2}$ lack dihydrouracil (DHU) arms, resulting in an inability to form a secondary structure. Similarly, *tRNA*$^{Ala}$ lacks a TΨC arm, rendering it incapable of forming a secondary structure. In contrast, the remaining tRNAs demonstrate a characteristic secondary structure reminiscent of a cloverleaf. A comparison with ancestral gastropods reveals distinct differences in three gene clusters (or genes), encompassing 15 tRNAs and eight PCGs. Notably, inversions and translocations represent the major types of rearrangements observed in *D. extinctorium*. Phylogenetic analysis demonstrates robust support for a monophyletic grouping of all Littorinimorpha species, with *D. extinctorium* representing a distinct Calyptraeoidea clade. In summary, this investigation provides the first complete mitochondrial dataset for a species of the Calyptraeidae, thus providing novel insights into the phylogenetic relationships within the Littorinimorpha.

## Introduction

Mitochondria are double—membrane—coated organelles found in most eukaryotes. Although most of a cell's DNA is contained in the nucleus, mitochondria have their own genome, known as the mitogenome. Attributable to its profoundly conserved characteristics, absence of extensive recombination, maternal inheritance, and elevated mutation rate [1–3], the

the National Center for Biotechnology Information, and the GenBank accession numbers of the mtgenome of D. extinctorium is OQ511529.

**Funding:** This research was financially supported by the Project of Bureau of Science and Technology of Zhoushan (No. 2021C21017), the National Key R&D Program of China (2019YFD0901204) and NSFC Projects of International Cooperation and Exchanges (42020104009). There was no additional external funding received for this study.

**Competing interests:** The authors have declared that no competing interests exist.

mitogenome has found extensive utility in the realms of comparative and evolutionary genomics [4], species identification, population genetics [5], molecular evolution, and phylogenetic relationships [6,7]. In particular, phylogeny based on complete mitochondrial genomes have demonstrated improved resolution compared to phylogenetic trees inferred from partial gene fragments such as *COI* and *16S rRNA* [8]. In recent years, mitochondrial genome sequencing and amplification techniques have rapidly developed, and mitochondrial genomes have been extensively utilized to reconstruct phylogenetic trees of different gastropods. For instance, Yang et al [9] sequenced the complete mitochondrial genomes of nine Nassariidae species and compared them with eight previously reported Nassariidae genomes, identifying the phylogenetic placement of these nine species within the gastropod clade. Genetic distance analysis and phylogenetic analysis both supported the distant relationship of *Nassarius jacksonianus* and *Nassarius acuticostus* to other *Nassarius* species. Furthermore, Yang et al [10] sequenced the complete mitochondrial genomes of two nassarids (Neogastropoda: Nassariidae: *Nassarius*), *Nassarius glans* and *Nassarius siquijorensis*, identifying the phylogenetic positions of these two species within *Nassarius*. In addition, Lee et al [11] reported the complete mitochondrial genome of *Semisulcospira gottschei* (Gastropoda: Caenogastropoda) and identified its phylogenetic relationship within Caenogastropoda. The study revealed that *Semisulcospira gottschei* is the closest relative to *Semisulcospira coreana*, and it was classified within the family Cerithioidea.

*Desmaulus extinctorium* is a marine snail that inhabits sandy substrates ranging from low intertidal to several metres subtidally. It belongs to the class Gastropoda, subclass Caenogastropoda, order Littorinimorpha, superfamily Calyptraeoidea, family Calyptraeidae, genus *Desmaulus*. *Desmaulus extinctorium* is abundant in southern China and Hongkong, with a widespread presence in the Indo-West Pacific region as well [12]. Previous research on this family has predominantly focused on morphology and growth [13–15]. Calyptraeid gastropods are known for their taxonomic challenges stemming from their simple, phenotypically variable shells [16]. As such, only a few studies have explored the phylogenetic analysis of this family. For instance, Cunha et al [17] conducted sequencing on a segment of the mitochondrial genome from the calyptraeoidean species *Calyptraea chinensis*, which belongs to the Littorinimorpha. Phylogenetic investigations have revealed that the Littorinimorpha does not form a monophyletic cluster. Meanwhile, Collin [18] examined how development modes influence the phylogeography and population dynamics of North Atlantic *Crepidula* (Gastropoda: Calyptraeidae). She created haplotype trees for each clade using 640 bp *COI* sequences. Examination of both the tree topology and AMOVA revealed that species undergoing direct development (hatching as benthic juveniles) displayed a more conspicuous population structure in comparison to those species undergoing planktonic development. Prior to our study, a complete mitochondrial genome of Calyptraeidae had not been uploaded to GenBank.

Littorinimorpha is a substantial order within Caenogastropoda (Class Gastropoda), encompassing 16 superfamilies according to the WoRMS database. Among marine snails, Caenogastropoda stands as the dominant group in terms of species numbers, diversity of habitats, ecological importance and behaviors. The current classification within Littorinimorpha was mainly established by Bouchet and Rocroi [19]. While Colgan et al [20] conducted an exhaustive phylogenetic investigation of Caenogastropoda, the interrelationships among families and superfamilies within the Caenogastropoda clade remain predominantly unresolved. The monophyly of both Littorinimorpha and Neogastropoda has been a topic of ongoing debate [21]. Cunha et al [22] conducted the sequencing of complete mitochondrial genomes for seven previously unanalyzed gastropod species. Subsequent phylogenetic analysis led to the rejection of the monophyletic status of Neogastropoda, attributed to the incorporation of Littorinimorpha lineages within this cluster. Additionally, Zhao et al. [23] sequenced the complete

mitochondrial genomes of intermediate host snails for *Schistosoma* and performed a phylogenetic analysis, revealing that neither Neogastopoda nor Littorinimorpha were monophyletic groups. Consequently, further research is necessary to refine the phylogenetic relationship within Caenogastropoda. Riedel [24] established the superfamily Ficoidea, separate from the Tonnoidea, but based on the sequencing of the complete mitochondrial genome of *Ficus variegata* Wang et al. [25] demonstrated that it fits within the Tonnoidea. And then, Jiang et al [26] reconstructed the phylogenetic tree of Littorinimorpha by sequencing the complete mitochondrial genome of two species in the Stromboidea. The findings provided evidence for the existence of three significant clades within Littorinimorpha: 1) Stromboidea, Tonnoidea, Littorinoidea, and Naticoidea, 2) Rissooidea alongside Truncatelloidea, and 3) Vermetoidea.

In this investigation, we have accomplished the comprehensive sequencing of the mitogenome for *D. extinctorium*. Furthermore, an elucidation of the gene structure within the mitogenome of *D. extinctorium* has been presented, coupled with a phylogenetic scrutiny encompassing 51 species from the Littorinimorpha taxon. This analysis is predicated upon the nucleotide sequences of 13 PCGs. As an outcome of this study, there has been an augmentation of the mitochondrial genome repertoire for Littorinimorpha, along with the provision of data requisite for subsequent phylogenetic assessments within the Littorinimorpha clade.

## Materials and methods

### Sampling and DNA extraction

We obtained a specimen of *D. extinctorium* from Ningde, Fujian Province, China (27˚04′812N, 120˚24′158″E). The initial morphological classification of these samples involved expert consultation with taxonomists at Zhejiang Ocean University's Marine Biological Museum. After collection, the specimen was rapidly submerged in absolute ethanol and stored at -20˚C. To confirm its classification, we relied on morphological traits, and we preserved fresh tissues in absolute ethanol before DNA extraction. We used the salt-extraction technique [27] to isolate complete genomic DNA, which was then stored at -20˚C.

### Genome sequencing, assembly and annotation

The mitogenomes of *D. extinctorium* were sequenced by Origin gene Co. Ltd., situated in Shanghai, China, employing the Illumina HiSeq X Ten sequencing platform. HiSeq X Ten libraries were prepared, incorporating an insert size ranging from 300 to 500 base pairs, sourced from genomic DNA samples. Each library yielded approximately 10 gigabases of raw data. Preprocessing procedures encompassed the elimination of low-quality reads, adapters, sequences containing high proportions of ambiguous bases ("N" bases), and those with a length below 25 base pairs. For assembly, the NOVOPlasty software [28] (accessible at https://github.com/ndierckx/NOVOPlasty) was utilized. Annotation and manual refinement of the assembly were performed with reference to established mitogenome datasets. De novo assembled mitogenomes were generated using MITOS tools [29] (accessed through the MITOS Web Server at uni-leipzig.de). Validation of sequence accuracy was achieved through alignment against mitochondrial genes of other Calyptraeoidea species, complemented by confirmation via the COI barcode sequence and NCBI BLAST searches [30].

Reads were reconstructed using a de novo assembly program, and subsequent annotation of complete mitogenomes was conducted using Sequin version 16.0. The mitogenome map of *D. extinctorium* was visualized utilizing the online tool Poksee (accessible at https://proksee.ca) [31]. Secondary structures of tRNA genes were forecasted and illustrated through the MITOS Web Server. To gain insights into coding sequence characteristics, relative synonymous codon usage (RSCU) values and substitution saturation for the 13 protein-coding genes (PCGs) were

computed utilizing DAMBE 5. Subsequent analysis of these values was executed using MEGA 7 [32]. Additionally, base compositional disparities and strand asymmetry among samples were assessed by evaluating GC-skews and AT-skews. These parameters were calculated using the following formulas: AT-skew = $[A-T]/[A+T]$ and GC skew = $[G-C]/[G+C]$. Substitution saturation for the 13 PCGs was quantified using DAMBE 5 [33].

## Gene order analysis

In addition to the mitogenomes sequenced in this study, we obtained an additional 51 complete mitogenomes of Littorinimorpha from GenBank (Table 1) for comparative analyses. The gene arrangements of all 51 mitogenomes were compared with the ancestral Gastropoda, with the aim of identifying potential novel gene orders that have not been reported in previous studies. To ensure that observed gene order differences were not caused by mis-annotations, any mitogenomes in Littorinimorpha that deviated from the ancestral pattern underwent re-annotation using MITOS [29].

## Phylogenetic analysis

Exploring the evolutionary relationships within the Littorinimorpha clade involved an analysis of 13 PCGs. These genes were sourced from a comprehensive dataset that included 51 complete mitogenome sequences. The mitogenome sequences were retrieved from the GenBank database (https://www.ncbi.nlm.nih.gov/genbank/). To provide additional context, two species from the Donacidae family were also included as representatives of the outgroup. The assessment of phylogenetic relationships utilized both Maximum Likelihood (ML) and Bayesian Inference (BI) methodologies [50–52].

The ML analysis, carried out with IQ-TREE 1.6.2, involved 1000 ultrafast likelihood bootstrap replicates. The choice of optimal models was guided by the Bayesian Information Criterion (BIC), leading to the adoption of the GTR + F + R6 model for each partition. We conducted Bayesian Inference (BI) analyses using the MrBayes 3.2 software, and model selection was facilitated by MrMTgui [53], a tool that connects PAUP, ModelTest, and MrModelTest across different platforms. For model selection, we chose the best-fit model (GTR + I + G) based on AIC results obtained from MrModelTest 2.3 [54]. Bayesian analyses were then performed in MrBayes, utilizing parameter values from either MrModelTest or ModelTest (nst = 6, rates = invgamma) [55]. The Bayesian analyses utilized Markov Chain Monte Carlo (MCMC) sampling, involving two independent runs of three hot chains and one cold chain. These chains ran simultaneously for 2,000,000 generations, with sampling intervals set at 1000 steps and a relative burn-in rate of 25%. We assessed the convergence of independent runs by examining the mean standard deviation of split frequencies ($< 0.01$). Finally, the resulting phylogenetic trees were visualized and edited using Figure Tree v.1.4.3 software [56].

## Results discussion

### Genome structure and composition

The complete mitogenome sequence of *D. extinctorium* constitutes a prototypical closed-circular molecule spanning 16,605 bp in length (GenBank accession number OQ511529). This genome encompasses a total of 37 genes, comprising 13 protein-coding genes (PCGs), 22 transfer RNAs (tRNAs), two ribosomal RNAs (*16S rRNA* and *12S rRNA*), and a concise noncoding region. This structural arrangement aligns consistently with the composition observed in the majority of previously investigated mollusks [57–59]. All these genes have been

**Table 1. List of species of Littorinimorpha analysed in this study and their GenBank accession numbers.**

| Superfamily | Family | Species | Accession no. | Size(bp) |
|---|---|---|---|---|
| Stromboidea | Strombidae | *Aliger gigas*[34] | MZ157283 | 15460 |
| | | *Conomurex luhuanus*[35] | NC_035726 | 15799 |
| | | *Harpago chiragra*[36] | MN885884 | 16404 |
| | | *Laevistrombus canarium*[37] | MT937083 | 15626 |
| | | *Lambis lambis*[36] | MH115428 | 15481 |
| | | *Strombus pugilis*[38] | MW244819 | 15809 |
| | | *Tridentarius dentatus*[38] | MW244820 | 15500 |
| | Aporrhaidae | *Aporrhais serresiana*[38] | MW244817 | 15455 |
| | Struthiolariidae | *Struthiolaria papulosa*[38] | MW244818 | 15475 |
| | Seraphsidae | *Terebellum terebellum*[38] | MW244821 | 15478 |
| | Rostellariidae | *Tibia fusus* | NC_065371 | 16083 |
| | | *Varicospira cancellata*[38] | MW244822 | 15864 |
| | Xenophoridae | *Xenophora japonica*[38] | MW244823 | 15684 |
| Truncatelloidea | Amnicolidae | *Baicalia turriformis*[39] | NC_035869 | 15127 |
| | | *Godlewskia godlewskii*[39] | NC_035870 | 15224 |
| | | *Maackia herderiana*[39] | NC_035871 | 15154 |
| | Pomatiopsidae | *Oncomelania hupensis* | NC_013073 | 15182 |
| | | *Tricula hortensis* | NC_013833 | 15179 |
| | Tateidae | *Potamopyrgus antipodarum*[40] | NC_070577 | 16846 |
| | | *Potamopyrgus estuarinus*[40] | NC_070576 | 16701 |
| Tonnoidea | Bursidae | *Bufonaria rana*[41] | MT408027 | 15510 |
| | Charoniidae | *Charonia lampas* | KU237290 | 15330 |
| | | *Charonia tritonis* | MT043269 | 15346 |
| | Cassidae | *Galeodea echinophora*[21] | NC_028003 | 15388 |
| | Cymatiidae | *Monoplex parthenopeus*[17] | NC_013247 | 15270 |
| Naticoidea | Naticidae | *Cryptonatica andoi*[42] | NC_046598 | 15302 |
| | | *Cryptonatica janthostoma*[42] | NC_046704 | 15235 |
| | | *Euspira gilva*[42] | NC_046593 | 15315 |
| | | *Euspira pila*[42] | NC_046703 | 15244 |
| | | *Glossaulax reiniana*[43] | NC_041162 | 15254 |
| | | *Mammilla mammata*[42] | NC_046597 | 15319 |
| | | *Mammilla kurodai*[42] | NC_046596 | 15309 |
| | | *Naticarius hebraeus*[21] | NC_028002 | 15384 |
| | | *Neverita didyma*[42] | NC_046594 | 15629 |
| | | *Notocochlis gualteriana*[42] | NC_046705 | 15176 |
| | | *Paratectonatica tigrina*[42] | NC_050661 | 15201 |
| | | *Polinices sagamiensis*[42] | NC_046595 | 15383 |
| | | *Tanea lineata*[42] | NC_050662 | 15156 |
| Cypraeoidea | Cypraeidae | *Cypraea tigris*[44] | MK783263 | 16177 |
| | | *Erronea errones* | NC_066082 | 15422 |
| Vermetoidea | Vermetidae | *Dendropoma gregarium*[45] | NC_014580 | 15641 |
| | | *Eualetes tulipa*[45] | NC_014585 | 15078 |
| | | *Thylacodes squamigerus*[45] | NC_014588 | 15544 |
| Ficoidea | Ficidae | *Ficus variegata* | NC_056153 | 15736 |
| Littorinoidea | Littorinidae | *Littoraria ardouiniana* | NC_066085 | 16261 |
| | | *Littoraria intermedia* | NC_064397 | 16194 |
| | | *Littoraria melanostoma* | NC_064398 | 16149 |

*(Continued)*

**Table 1.** (Continued)

| Superfamily | Family | Species | Accession no. | Size(bp) |
|---|---|---|---|---|
| | | *Littoraria sinensis*[46] | MN496138 | 16420 |
| | | *Littorina brevicula*[47] | NC_050987 | 16356 |
| | | *Littorina saxatilis* | NC_030595 | 16887 |
| | | *Melarhaphe neritoides*[48] | MH119311 | 15676 |
| Calyptraeoidea | Calyptraeidae | *Desmaulus extinctorium* | OQ511529 | 16572 |
| Outgroup | | *Donax variegatus*[49] | NC_035986 | 17195 |
| | | *Donax vittatus*[49] | NC_035987 | 17070 |

discerned and are depicted in Fig 1 and Table 2. Among the 37 genes, the majority are localized on the heavy (H-) strand, except for eight tRNAs (*tRNA-Phe*, *His*, *Pro*, *Leu*, *Val*, *Gln*, *Cys*, and *Tyr*). (Fig 1. Maps of the mitochondrial genomes of *D. extinctorium*.)

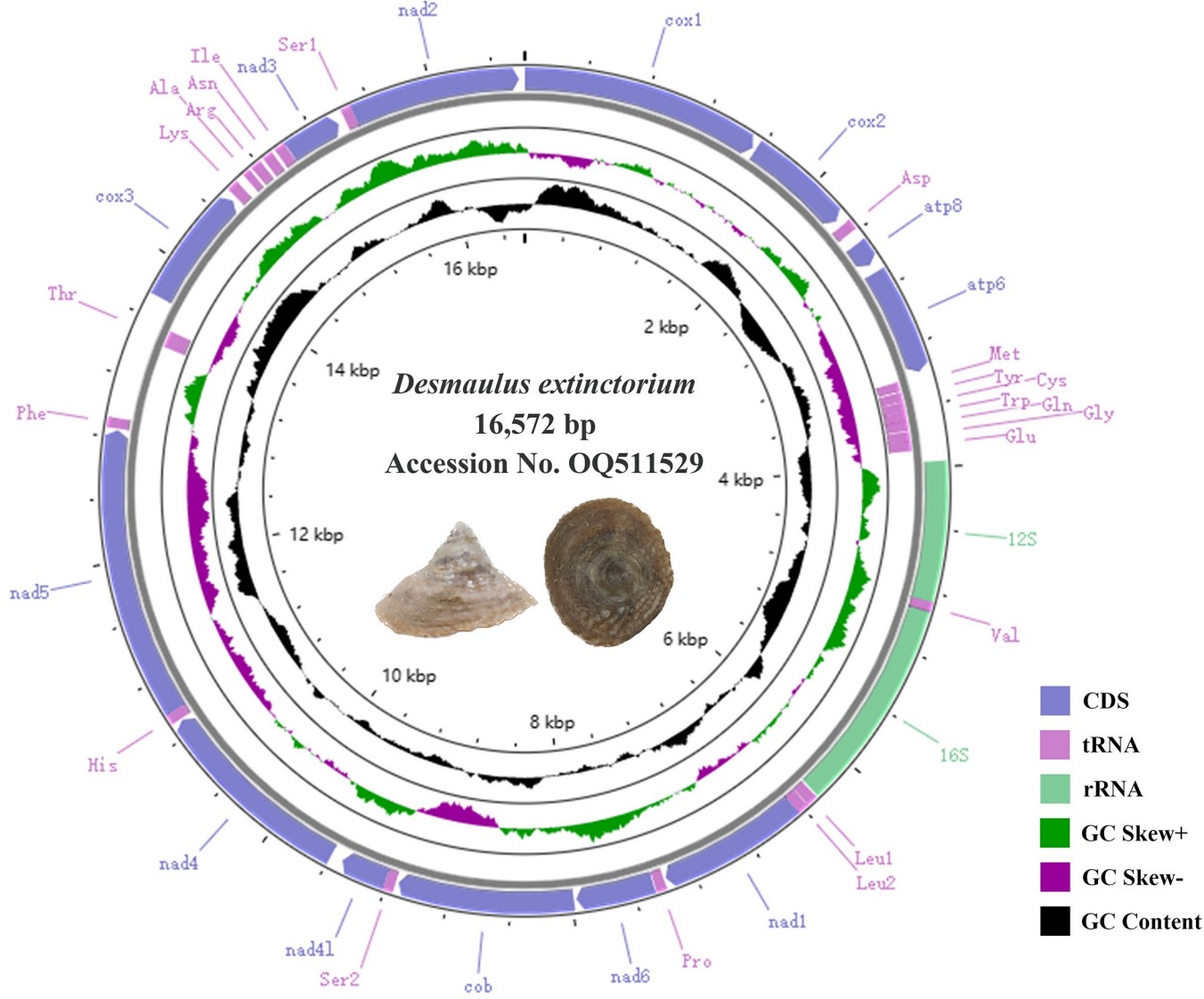

**Fig 1. Maps of the mitochondrial genomes of *D. extinctorium*.**

**Table 2. Mitochondrial genome organization of *D. extinctorium*.**

| Gene | Direction | Position | | Length/bp | Start/Stop codon | Intergenic Nucleotide(bp) | Anticodon |
|---|---|---|---|---|---|---|---|
| COX1 | H | 1 | 1551 | 1551 | ATT/TAA | 22 | |
| COX2 | H | 1574 | 2266 | 693 | ATG/TAA | 46 | |
| tRNA$^{Asp}$ | H | 2313 | 2382 | 70 | | 82 | GTC |
| ATP8 | H | 2465 | 2623 | 159 | ATG/TAA | 57 | |
| ATP6 | H | 2681 | 3376 | 696 | ATG/TAA | 26 | |
| tRNA$^{Met}$ | L | 3403 | 3468 | 66 | | 8 | CAT |
| tRNA$^{Tyr}$ | L | 3477 | 3542 | 66 | | 4 | GTA |
| tRNA$^{Cys}$ | L | 3547 | 3612 | 66 | | 0 | GCA |
| tRNA$^{Trp}$ | L | 3613 | 3679 | 67 | | -2 | TCA |
| tRNA$^{Gln}$ | L | 3678 | 3743 | 66 | | 4 | TTG |
| tRNA$^{Gly}$ | L | 3748 | 3813 | 66 | | -1 | TCC |
| tRNA$^{Glu}$ | L | 3813 | 3882 | 70 | | 80 | TTC |
| 12S rRNA | H | 3963 | 4858 | 896 | | -1 | |
| tRNA$^{Val}$ | H | 4858 | 4925 | 68 | | -10 | TAC |
| 16S rRNA | H | 4916 | 6279 | 1364 | | 13 | |
| tRNA$^{Leu1}$ | H | 6293 | 6364 | 72 | | 4 | TAG |
| tRNA$^{Leu2}$ | H | 6369 | 6438 | 70 | | 0 | TAA |
| NAD1 | H | 6439 | 7383 | 945 | ATG/TAA | 12 | |
| tRNA$^{Pro}$ | H | 7396 | 7463 | 68 | | 6 | TGG |
| NAD6 | H | 7470 | 7973 | 504 | ATG/TAA | 16 | |
| Cytb | H | 7990 | 9129 | 1140 | ATG/TAA | 17 | |
| tRNA$^{Ser2}$ | H | 9147 | 9212 | 66 | | 0 | TGA |
| NAD4l | H | 9213 | 9515 | 303 | ATG/TAG | 80 | |
| NAD4 | H | 9596 | 10900 | 1305 | ATG/TAA | 10 | |
| tRNA$^{His}$ | H | 10911 | 10976 | 66 | | 0 | GTG |
| NAD5 | H | 10977 | 12848 | 1872 | ATG/TAG | 10 | |
| tRNA$^{Phe}$ | H | 12859 | 12924 | 66 | | 12 | GAA |
| tRNA$^{Thr}$ | L | 13573 | 13640 | 68 | | 104 | TGT |
| COX3 | H | 13745 | 14524 | 780 | ATG/TAA | 29 | |
| tRNA$^{Lys}$ | H | 14554 | 14628 | 75 | | 13 | TTT |
| tRNA$^{Ala}$ | H | 14684 | 14734 | 51 | | 17 | TGC |
| tRNA$^{Arg}$ | H | 14752 | 14821 | 70 | | 21 | TCG |
| tRNA$^{Asn}$ | H | 14843 | 14912 | 70 | | 23 | GTT |
| tRNA$^{Ile}$ | H | 14936 | 15005 | 70 | | 0 | GAT |
| NAD3 | H | 15010 | 15366 | 357 | ATG/TAG | 1 | |
| tRNA$^{Ser1}$ | H | 15416 | 15483 | 68 | | 2 | GCT |
| NAD2 | H | 15484 | 16572 | 1089 | ATG/TAA | 36 | |

The longest gene, *ND5*, stretches across 1872 base pairs, whereas the shortest is *tRNA$^{Ala}$*, comprising a mere 51 base pairs. The *D. extinctorium* mitogenome comprises four regions displaying overlap. Of these, one involves a 10 bp overlap with *tRNA$^{Val}$*, and the remaining three exhibit overlaps shorter than 10 bp with *tRNA$^{Trp}$* (2 bp), *tRNA$^{Gly}$* (1 bp), and *16S rRNA* (1 bp). Additionally, the *D. extinctorium* mitogenome accommodates 1393 bp of intergenic spacers distributed across 28 regions, ranging in size from 3 to 648 bp (Table 2).

Regarding nucleotide composition, the *D. extinctorium* mitogenome is comprised of A (27.73%), T (42.47%), G (18.08%), and C (11.71%), demonstrating a conspicuous AT bias. These findings parallel not only those observed in numerous mollusks [60,61] but also in

certain crustaceans like crabs and lobsters [62,63]. The cumulative A + T (%) content of the mitogenomes stands at 70.20%. Calculated for the selected complete mitogenomes, the AT-skew of the *D. extinctorium* mitogenome is negative (−0.210), while the GC-skew is positive (0.214), implying a higher abundance of Ts and Cs than As and Gs. These outcomes align with those identified in specific Neritidae species [57].

## Transfer RNAs, ribosomal RNAs

Similar to the prevailing pattern in many invertebrate species [64,65], the mitogenome of *D. extinctorium* harbors a total of 22 tRNA genes. Among these, fourteen are encoded by the heavy strand (H-), while the remaining ones are encoded by the light strand (L-). Across the entire mitogenome, the size of tRNA molecules spans from 51 to 75 bp, collectively encompassing a length of 1485 bp, characterized by a pronounced AT bias (70.23%). The AT-skew and GC-skew values are recorded as– 0.014 and 0.158, respectively, signifying a subtle inclination towards adenine usage and a conspicuous predilection for guanine usage (Table 3). The $tRNA^{Ser1}$ and $tRNA^{Ser2}$, due to the absence of dihydrouracil (DHU) arms, along with $tRNA^{Ala}$, due to the lack of a TΨC arm, are unable to adopt a secondary structure. Conversely, other tRNAs possess the capacity to fold into a conventional clover-leaf secondary structure. Notably, the structural variation observed in $tRNA^{Ser1}$ corresponds with the $tRNA^{Ser1}$ configuration documented in other invertebrate mitogenomes [66]. Moreover, G-C mismatches are evident in all tRNAs except $tRNA^{Leu2}$, $tRNA^{Met}$, $tRNA^{Trp}$, and $tRNA^{Tyr}$. (Fig 2. Secondary structure of the tRNA genes in the mitogenome of *D. extinctorium*. The tRNAs are labeled with the abbreviations of their corresponding amino acids. Blue dots indicate normal conditions and yellow dots indicate base mismatches.).

The sizes of the *12S rRNA* and *16S rRNA* components are 896 bp and 1364 bp, respectively, typically demarcated by $tRNA^{Val}$ (Table 2). These dimensions align comparably with those observed in other invertebrate species. The A-T content of the rRNAs is determined to be 73.05%. AT-skew and GC-skew values are recorded as– 0.044 and 0.268, respectively, indicating a modest tendency towards adenine utilization and a marked preference for guanine

**Table 3. Nucleotide contents of the coding and non-coding regions of the mitochondrial genome of *D. extinctorium*, indicating AT-, GC-skew ratios.**

| Region | Size(bp) | A (%) | T (%) | G (%) | C (%) | A+T (%) | AT-skew | GC-skew |
|---|---|---|---|---|---|---|---|---|
| Mitogenome | 16608 | 27.73 | 42.47 | 18.08 | 11.72 | 70.20 | -0.210 | 0.213 |
| COX1 | 1551 | 23.92 | 43.13 | 20.05 | 12.89 | 67.05 | -0.287 | 0.217 |
| COX2 | 693 | 27.13 | 38.96 | 20.49 | 13.42 | 66.09 | -0.179 | 0.209 |
| ATP8 | 159 | 27.67 | 44.65 | 15.72 | 11.95 | 72.32 | -0.235 | 0.136 |
| ATP6 | 696 | 23.13 | 46.70 | 16.67 | 13.51 | 69.83 | -0.337 | 0.105 |
| COX3 | 780 | 21.67 | 42.31 | 22.31 | 13.72 | 63.98 | -0.323 | 0.238 |
| NAD3 | 357 | 19.89 | 47.90 | 20.73 | 11.48 | 67.79 | -0.413 | 0.287 |
| NAD1 | 945 | 24.02 | 45.29 | 18.20 | 12.49 | 69.31 | -0.307 | 0.186 |
| NAD5 | 1872 | 26.82 | 42.63 | 16.61 | 13.94 | 69.45 | -0.228 | 0.087 |
| NAD4 | 1305 | 25.21 | 46.13 | 16.86 | 11.80 | 71.34 | -0.293 | 0.176 |
| NAD4l | 303 | 26.73 | 43.89 | 19.80 | 9.57 | 70.62 | -0.243 | 0.348 |
| NAD6 | 504 | 26.39 | 46.43 | 18.65 | 8.53 | 72.82 | -0.275 | 0.372 |
| Cytb | 1140 | 24.39 | 44.39 | 17.81 | 13.42 | 68.78 | -0.291 | 0.140 |
| NAD2 | 1089 | 25.34 | 46.37 | 18.64 | 9.64 | 71.71 | -0.293 | 0.318 |
| tRNAs | 1485 | 34.61 | 35.62 | 17.24 | 12.53 | 70.23 | -0.014 | 0.158 |
| rRNAs | 2260 | 34.91 | 38.14 | 17.08 | 9.87 | 73.05 | -0.044 | 0.268 |
| PCGs | 11394 | 24.84 | 44.25 | 18.47 | 12.44 | 69.09 | -0.281 | 0.195 |

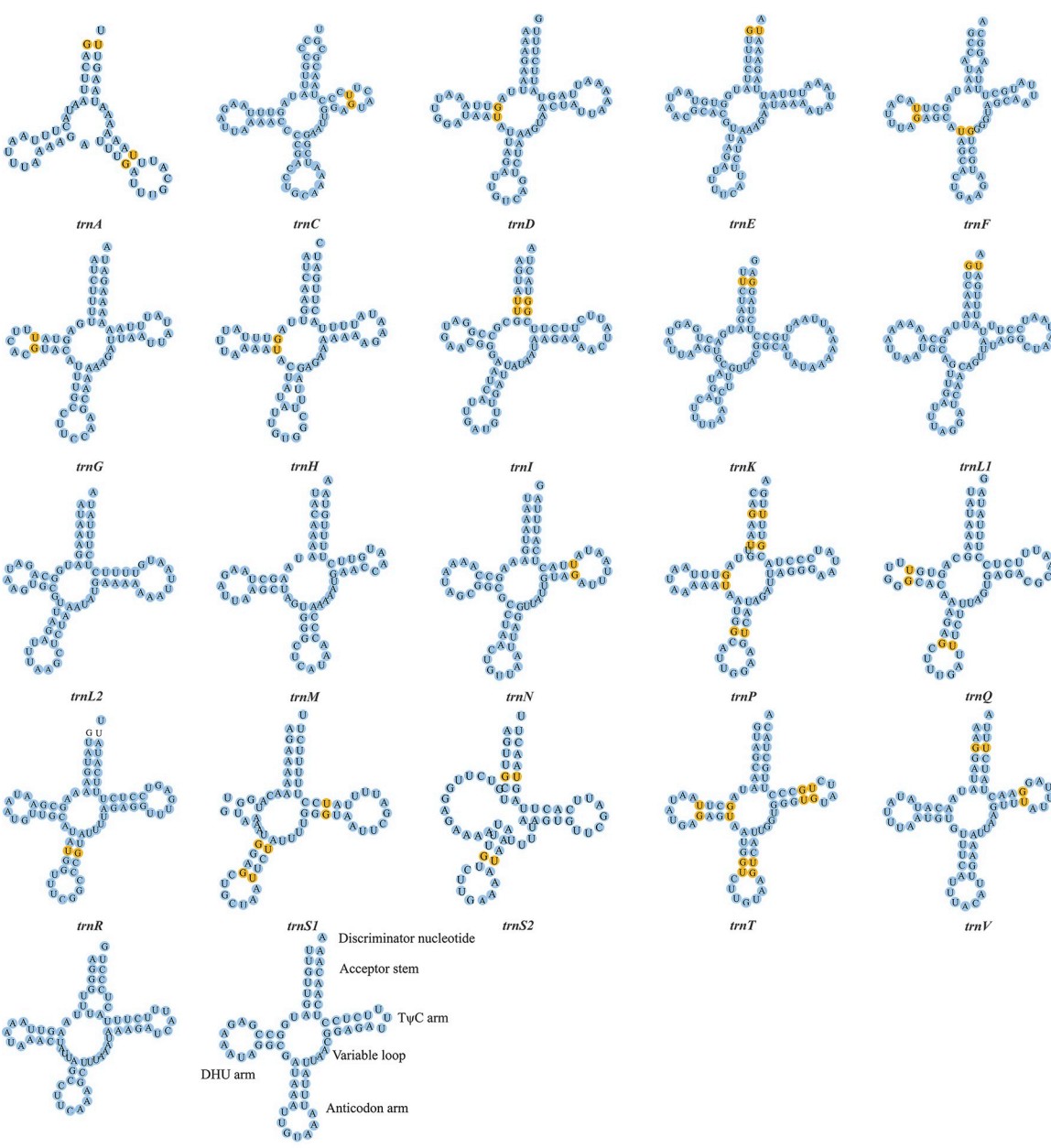

**Fig 2. Secondary structure of the tRNA genes in the mitogenome of *D. extinctorium*.** The tRNAs are labeled with the abbreviations of their corresponding amino acids. Blue dots indicate normal conditions and yellow dots indicate base mismatches.

utilization (Table 3). The control region (CR), positioned between $tRNA^{Thr}$ and $tRNA^{Phe}$, spans a length of 648 bp.

## PCGs and codon usage

The result presents the initiation and termination codons for all Protein-Coding Genes (PCGs) within *D. extinctorium* in Table 3. The mitochondrial genome of D. extinctorium encompasses a total of 13 PCGs, comprising a cytochrome b (*Cytb*), two ATPases (*ATP6* and *ATP8*), three cytochrome oxidases (*COI–III*), and seven NADH dehydrogenases (*ND1–6* and

**Table 4. Relative synonymous codon usage (RSCU) in the mitogenomes of *D. extinctorium*.**

| Codon | Count | RSCU | Codon | Count | Codon | Codon | Count | RSCU | Codon | Count | Codon |
|---|---|---|---|---|---|---|---|---|---|---|---|
| GCU(A) | 84.0 | 2.11 | CCU(P) | 64.0 | 2.08 | AGA(S) | 85.0 | 1.17 | CAU(H) | 57.0 | 1.54 |
| GCC(A) | 19.0 | 0.48 | CCC(P) | 19.0 | 0.62 | AGG(S) | 83.0 | 1.14 | CAC(H) | 17.0 | 0.46 |
| GCA(A) | 41.0 | 1.03 | CCA(P) | 27.0 | 0.88 | AUU(I) | 293.0 | 1.70 | ACU(T) | 97.0 | 2.38 |
| GCG(A) | 15.0 | 0.38 | CCG(P) | 13.0 | 0.42 | AUC(I) | 52.0 | 0.30 | ACC(T) | 20.0 | 0.49 |
| UGU(C) | 123.0 | 1.43 | CAA(Q) | 53.0 | 1.15 | AAA(K) | 179.0 | 1.39 | ACA(T) | 34.0 | 0.83 |
| UGC(C) | 49.0 | 0.57 | CAG(Q) | 39.0 | 0.85 | AAG(K) | 79.0 | 0.61 | ACG(T) | 12.0 | 0.29 |
| GAU(D) | 112.0 | 1.75 | CGU(R) | 35.0 | 1.77 | UUA(L) | 336.0 | 2.66 | GUU(V) | 182.0 | 2.25 |
| GAC(D) | 16.0 | 0.25 | CGC(R) | 5.0 | 0.25 | UUG(L) | 160.0 | 1.27 | GUC(V) | 33.0 | 0.41 |
| GAA(E) | 95.0 | 1.43 | CGA(R) | 18.0 | 0.91 | CUU(L) | 126.0 | 1.00 | GUA(V) | 66.0 | 0.82 |
| GAG(E) | 38.0 | 0.57 | CGG(R) | 21.0 | 1.06 | CUC(L) | 26.0 | 0.21 | GUG(V) | 42.0 | 0.52 |
| UUU(F) | 512.0 | 1.65 | UCU(S) | 113.0 | 1.55 | CUA(L) | 67.0 | 0.53 | UGA(W) | 115.0 | 1.11 |
| UUC(F) | 108.0 | 0.35 | UCC(S) | 44.0 | 0.60 | CUG(L) | 43.0 | 0.34 | UGG(W) | 92.0 | 0.89 |
| GGU(G) | 102.0 | 1.47 | UCA(S) | 83.0 | 1.14 | AUA(M) | 135.0 | 1.19 | UAU(Y) | 240.0 | 1.56 |
| GGC(G) | 41.0 | 0.59 | UCG(S) | 31.0 | 0.43 | AUG(M) | 91.0 | 0.81 | UAC(Y) | 68.0 | 0.44 |
| GGA(G) | 65.0 | 0.94 | AGU(S) | 100.0 | 1.37 | AAU(N) | 189.0 | 1.65 | UAA(*) | 174.0 | 1.25 |
| GGG(G) | 70.0 | 1.01 | AGC(S) | 43.0 | 0.59 | AAC(N) | 40.0 | 0.35 | UAG(*) | 105.0 | 0.75 |

*ND4L*). This configuration aligns with the established structural pattern observed in the Muricidae family [64]. The collective length of these 13 PCGs amounts to 11,484 bp. Within this set, the individual PCGs exhibit a range of lengths spanning from 159 to 1,872 bp. Notably, the average A+T content stands at 69.13%, with variations across the spectrum from 63.98% (*COIII*) to 72.82% (*ND6*) (Table 2). The AT-skew and GC-skew values are calculated as -0.281 and 0.195, respectively (Table 4). It is noteworthy that all PCGs commence with the initiation codon ATG, except for *ND4*, which employs ATT as its start codon. Furthermore, the majority of PCGs terminate with TAA, whereas *ND4L*, *ND5*, and *ND5* employ TAG as their respective stop codons (Table 4). Examining the amino acid utilization in *D. extinctorium*, tRNA$^{Phe}$ emerges as the most frequently employed, while *tRNA$^{His}$* is the least prevalent (Fig 2). Relative synonymous codon usage (RSCU) values for the 13 PCGs in *D. extinctorium* are presented in Table 4 and Fig 3. Among these, UUA (Leu) ranks as the most frequently encountered codon, whereas CUC (Leu) stands as the least common codon. (Fig 3. Codon usage patterns in the mitogenome of *D. extinctorium*. CDspT, codons per thousand codons. Codon families are provided on the x-axis (A) and the relative synonymous codon usage (RSCU) (B)).

## Gene re-arrangement

Rearrangements in mitochondrial gene order present an autonomous dataset for resolving evolutionary relationships. Shared patterns of mitogenome gene order rearrangements among distinct taxonomic groups are likely indicative of common ancestry rather than products of convergent evolution [6,67]. In comparison to the ancestral gastropod gene arrangement, significant rearrangements are evident in the mitogenome of *D. extinctorium*. As illustrated in Fig 4, a minimum of three gene clusters (or genes) differ notably from the conventional arrangement, encompassing 15 tRNA genes (*M, Y, C, W, Q, G, E, V, L, P, S, H, F*, and *T*), as well as eight protein-coding genes (*16S rRNA, 12S rRNA, NAD1, NAD6, Cytb, NAD4L, NAD4,* and *NAD5*). The rearrangement of these three gene clusters (or genes) is detailed as follows (Fig 4): (1) The *M-Y-C-W-Q-G-E* cluster has relocated downstream of *ATP6*; (2) The *T* cluster has shifted downstream of *F*; (3) The *F-ND5-H-ND4-ND4L-S-Cytb-ND6-P-ND1-L-16S-V-12S* underwent inversion and translocation. (Fig 4. Comparison of mitochondrial gene

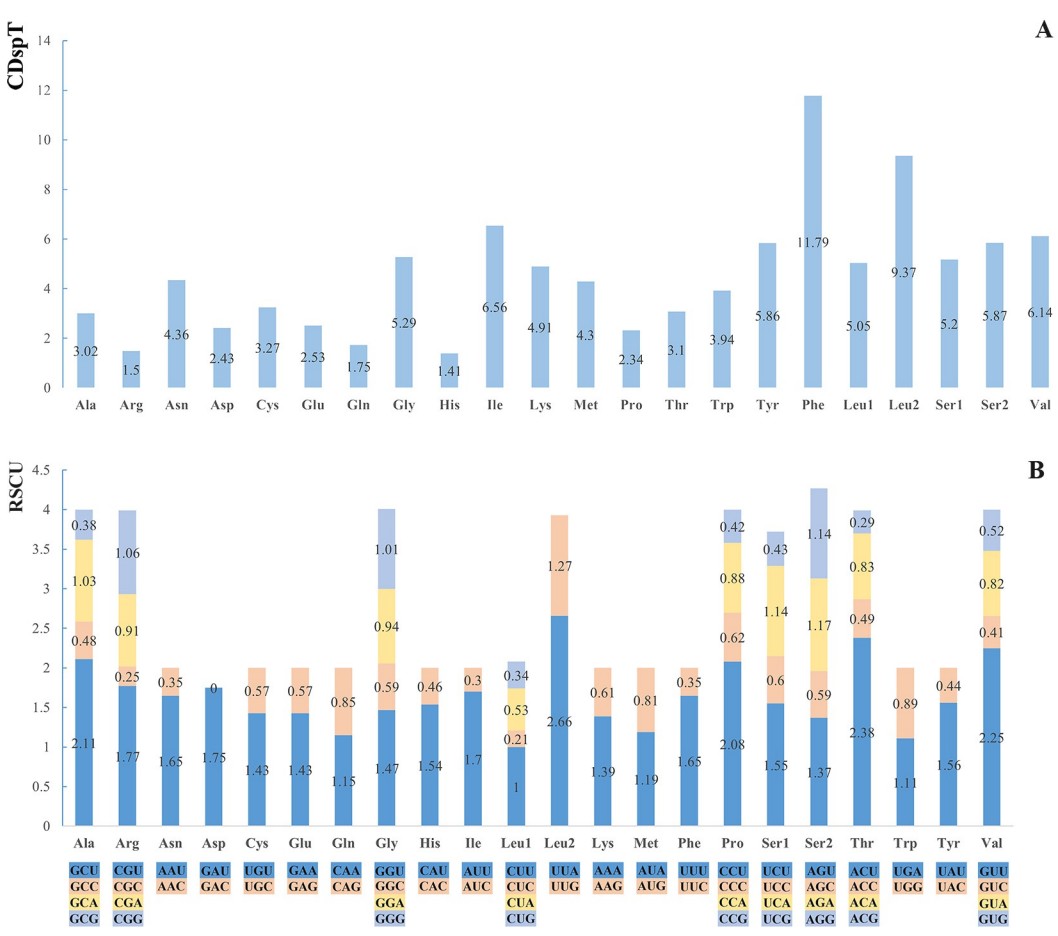

**Fig 3. Codon usage patterns in the mitogenome of *D. extinctorium*. CDspT, codons per thousand codons.** Codon families are provided on the x-axis (A) and the relative synonymous codon usage (RSCU) (B).

rearrangements of the *D. extinctorium*. The green squares represent PCGs, the yellow squares represent tRNAs, and the orange squares represent rRNAs. The position at the top indicates that it is encoded in the H chain, and the position at the bottom indicates that it is encoded in the L chain.)

Furthermore, mitochondrial gene rearrangements have frequently been linked to heightened rates of evolution [68]. Prior investigations have identified a notable positive correlation in mitochondrial genomes between rates of gene order rearrangement and accelerated evolutionary rates [69]. Intriguingly, when compared to the extensive gene rearrangements observed in Lottiidae, Littorinimorpha exhibits minimal differences in genetic order, with the exception of Vermetoidea [70]. We postulate that this circumstance could be attributed to the relatively modest variations in genome size among Littorinimorpha species, ranging from 15,127 bp to 17,195 bp (Tab 1), while the mitochondrial genome size within Lottiidae spans from 16,319 bp to 26,835 bp. Further investigations are warranted to scrutinize this association within a broader spectrum of Gastropoda groups.

In the context of gene rearrangement patterns, three primary categories are recognized [71]: (1) shuffling, where genes migrate from their original locations to adjacent positions on the same strand, typically without traversing protein-coding genes; (2) translocation, in which

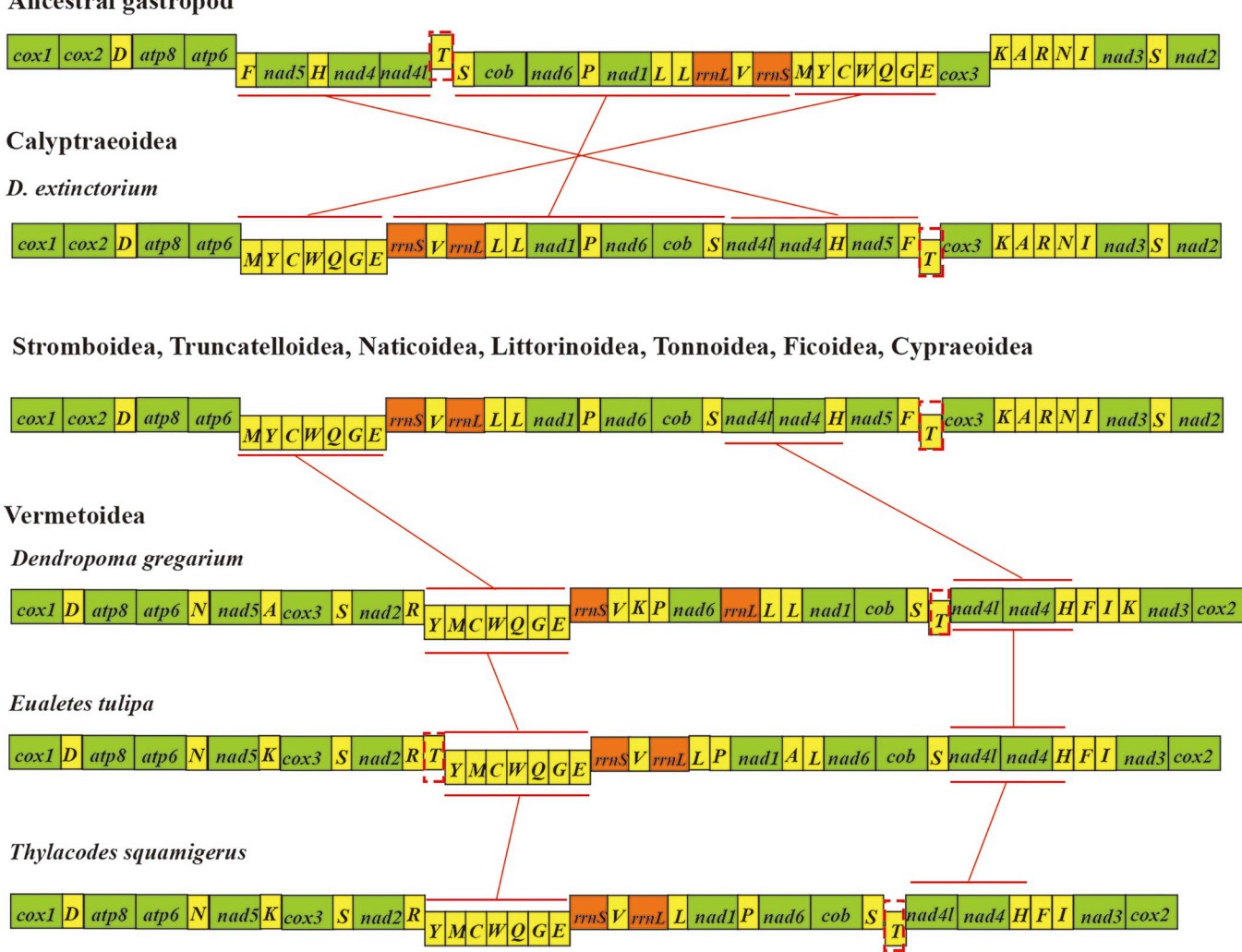

**Fig 4. Comparison of mitochondrial gene rearrangements of the *D. extinctorium*.** The green squares represent PCGs, the yellow squares represent tRNAs, and the orange squares represent rRNAs. The position at the top indicates that it is encoded in the H chain, and the position at the bottom indicates that it is encoded in the L chain.

genes traverse several genes, often including protein-coding genes, relocating from their original positions to new sites; (3) inversion, involving the switch of genes from one strand to the other. Based on the characteristics of mitochondrial sequences, our analysis suggests that inversion and translocation are the predominant types of rearrangements observed in *D. extinctorium*.

Furthermore, we conducted a comparative examination of the gene order in *D. extinctorium* against other superfamilies within Littorinimorpha. With the exception of Vermetoidea, the gene order across other superfamilies remains largely consistent. Notably, in Vermetoidea, significant deviations in gene order primarily pertain to tRNAs. In addition, the *M-Y-C-W-Q-G-E* cluster within the mitochondrial genome of Vermetoidea has undergone inversion, a phenomenon observed in other gastropod mitochondrial genomes [72], resulting in disruption and rearrangement. Intriguingly, a remarkably similar set of genes undergoes rearrangement in the common ancestor of Caenogastropoda, although the integrity of the *M-Y-C-W-Q-G-E* cluster is maintained [22,73,74]. These findings align with the conclusions

drawn from gene order-based phylogenetic analysis, underscoring the utility of comparing mitochondrial gene rearrangements as a valuable tool in phylogenetic studies.

## Phylogenetic relationships

In this current study, we conducted an analysis of phylogenetic relationships using the sequences of 13 protein-coding genes (PCGs). The primary objective was to gain insights into the interrelationships within the Littorinimorpha clade, focusing on *D. extinctorium*. Additionally, we included 51 other well-known Littorinimorpha species in our analysis, with *Donax variegatus* and *Donax vittatus* serving as outgroups. Both the Maximum Likelihood (ML) tree and the Bayesian Inference (BI) tree revealed consistent topological structures, although they exhibited varying degrees of support values. Notably, BI generally yielded higher support values, with most nodes having a support value of 1. In contrast, the support values in ML, except for three nodes in the Stromboidea superfamily, were below 70, and the majority of other branches had support values above 90. Consequently, we present and display only one topology (ML) with both support values. (Fig 5. The phylogenetic tree was inferred from the nucleotide sequences of 13 mitogenome PCGs using BI and ML methods. Numbers on branches indicate posterior probability (BI) and bootstrap support (ML)).

Among the 19 families encompassed within our phylogenetic tree, each individual family constitutes a monophyletic clade, bolstered by elevated nodal support values. Phylogenetic analysis showed that nine superfamilies within the Littorinimorpha show the following relationship: ((((((Naticoidea + Littorinoidea) + Truncatelloidea) + Tonnoidea + Ficoidea + Cypraeoidea) +Stromboidea) + Calyptraeoidea) + Vermetoidea), and all nine of them are monophyletic groups, some previous studies have shown that this is plausible [75,76], and Naticoidea and Littorinoidea are the closest sisters to each other. Additionally, phylogenetic tree showed that (Tonnoidea + Ficoidea + Cypraeoidea) formed a clade which showing were sister groups in this tree, while *D. extinctoriu*m alone forms a Calyptraeoidea clade, and (Calyptraeoidea + (Stromboidea + (Tonnoidea + Ficoidea + Cypraeoidea) + Truncatelloidea + (Naticoidea + Littorinoidea))) formed a clade. Vermetoidea is placed at the basal position of the monophyletic Littorinimorpha, this is consistent with previous research [73], and this can also be related to the results of gene re-arrangements, only the Vermetoidea has a significantly different genetic sequence from the rest of the species, so Vermetoidea is at the bottom of the phylogenetic tree. Stromboidea is the superfamily containing the largest number of families, it is a highly diverse group. Stromboidea is currently understood to comprise six extant families: Aporrhaidae, Rostellariidae, Seraphsidae, Strombidae, Struthiolariidae and Xenophoridae. Within each superfamily, each family forms a distinct clade. The results of phylogenetic relationships in the superfamil were consistent with the findings of Irwin et al [38]. In our study, Naticidae is the family of which most species have been included, representing a large number of genera. The Littorinidae are its sister group, of which a substantial number of species has been included in our study, however only representing a selection of the genera.

## Conclusion

In this investigation, we conducted the sequencing of the mitogenome of *D. extinctorium* employing next-generation sequencing techniques, thus yielding novel mitochondrial data pertinent to Calyptraeidae. An exhaustive examination of the mitogenome of *D. extinctorium* revealed its substantial resemblance to other representatives of the Littorinimorpha order, characterized by several notable features, including AT-skew and a codon usage bias, among others. Comparative analysis with the ancestral gastropod indicated a noteworthy rearrangement in the gene order of the *D. extinctorium* mitogenome. The Littorinimorpha exhibited

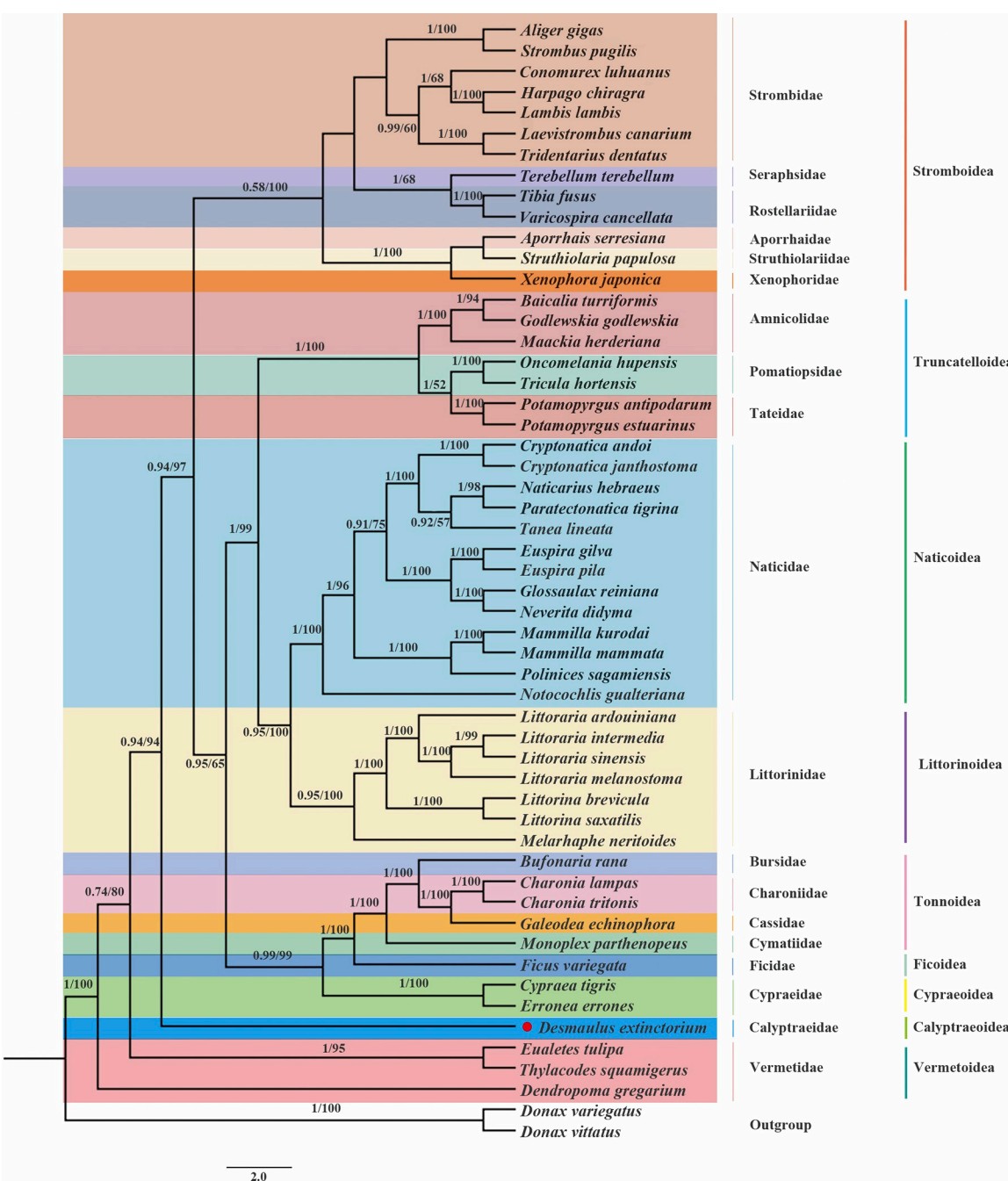

**Fig 5. The phylogenetic tree was inferred from the nucleotide sequences of 13 mitogenome PCGs using BI and ML methods.**
Numbers on branches indicate posterior probability (BI) and bootstrap support (ML).

four distinct rearrangement patterns, with their rearrangement similarity consistently mirroring their phylogenetic relationships. Our phylogenetic tree displayed both congruities and disparities when compared to preceding studies. Phylogenetic analyses indicated the formation of an exclusive Calyptraeoidea clade by *D. extinctorium*, whereas (Calyptraeoidea + (Stromboidea + (Tonnoidea + Ficoidea + Cypraeoidea) + Truncatelloidea + (Naticoidea + Littorinoidea))) constituted a distinct clade. Despite a limited number of species available for a robust

phylogenetic analysis, our phylogeny garnered statistical support and aspires to provide a rational framework for future phylogenetic inquiries within the realm of Calyptraeoidea. These findings not only offer insights into the gene arrangement characteristics within Littorinimorpha mitogenomes but also establish the groundwork for further explorations into the phylogenetic aspects of Littorinimorpha.

## Author Contributions

**Conceptualization:** Yanwen Ma, Yingying Ye.

**Data curation:** Yanwen Ma, Jiji Li.

**Formal analysis:** Yingying Ye.

**Funding acquisition:** Yingying Ye.

**Investigation:** Biqi Zheng, Wei Meng.

**Methodology:** Yanwen Ma.

**Project administration:** Biqi Zheng, Yingying Ye.

**Resources:** Biqi Zheng.

**Software:** Yanwen Ma.

**Supervision:** Kaida Xu.

**Validation:** Kaida Xu.

**Visualization:** Yanwen Ma.

**Writing – original draft:** Yanwen Ma.

**Writing – review & editing:** Jiji Li, Yingying Ye.

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
