## [Decision Letter · Decision Letter 0]

23 Jan 2024

PONE-D-23-34396Characterization of the complete mitochondrial genome of Desmaulus extinctorium (Littorinimorpha, Calyptraeoidea, Calyptraeidae) and molecular phylogeny of LittorinimorphaPLOS ONE

Dear Dr. Ye,

Thank you for submitting your manuscript to PLOS ONE. After careful consideration, we feel that it has merit but does not fully meet PLOS ONE’s publication criteria as it currently stands. Therefore, we invite you to submit a revised version of the manuscript that addresses the points raised during the review process.

**Both reviewers of the paper indicated that the data, study, and results are valuable and should be published. However, they also had some concerns about language, grammar etc. In addition, one reviewer has some more substantial concerns. I expect that all of these concerns can be addressed and look forward to your revised manuscript.**

We look forward to receiving your revised manuscript.

Kind regards,

Sean Michael Prager, Ph.D.

Academic Editor

PLOS ONE

Journal Requirements:

This research was financially supported by the National Key R&D Program of China (2019YFD0901204), NSFC Projects of International Cooperation and Exchanges (42020104009).

3. For studies involving third-party data, we encourage authors to share any data specific to their analyses that they can legally distribute. PLOS recognizes, however, that authors may be using third-party data they do not have the rights to share. When third-party data cannot be publicly shared, authors must provide all information necessary for interested researchers to apply to gain access to the data. (https://journals.plos.org/plosone/s/data-availability#loc-acceptable-data-access-restrictions) 

a) A description of the data set and the third-party source

b) If applicable, verification of permission to use the data set

c) Confirmation of whether the authors received any special privileges in accessing the data that other researchers would not have

d) All necessary contact information others would need to apply to gain access to the data

Reviewers' comments:

Reviewer's Responses to Questions

**Comments to the Author**

1. Is the manuscript technically sound, and do the data support the conclusions?

Reviewer #1: Yes

Reviewer #2: Partly

2. Has the statistical analysis been performed appropriately and rigorously? 

Reviewer #1: I Don't Know

Reviewer #2: Yes

3. Have the authors made all data underlying the findings in their manuscript fully available?

Reviewer #1: Yes

Reviewer #2: Yes

4. Is the manuscript presented in an intelligible fashion and written in standard English?

Reviewer #1: No

Reviewer #2: No

5. Review Comments to the Author

Reviewer #1: The results presented are clear, and the phylogenetic tree is well supported. This manuscript includes far more descriptive details of the mitogenome than other publications of individual mitogenomes.

The MS could be shortened and the language improved.

Reviewer #2: Summary

The core of the paper is the provision of complete mitochondrial genome of Desmaulus extinctorium. Those data have been provided. These new data confirm that the Calyptraeoidea form part of the Littorinimorpha clade. Interestingly, the Calyptraeoidea separated early from all other Littorinimorpha included in the study. Regrettably, data on other Calyptraeoidea or the possibly closely related Capuloidea and Hipponicoidea have not been included.

[Note the detailed molecular analysis has not been reviewed by me]

Examples and evidence

Major issues

• A major risk in molecular studies is misidentification of the specimens that were sequenced (there are numerous examples). Therefore it is useful to add photographs of those specimens and indicate where they were collected. As many species included are based on previous or ongoing studies, can you provide the relevant literature references for each? [By the way, I see other papers, e.g. in respectable magazines like Nature, that provide less data than the current study].

• Add at least photographs of the specimen analysed.

• Line 49 I have no idea what geological mud is, but suppose you mean siliciclastic mud (instead of carbonate mud) . Anyway, Desmaulus extinctorium is typically found on sandy substrates from low intertidal to several metres subtidally. It is a commensal of hermit crabs, living on the outside of gastropod shells inhabited by these crabs. [own observations and e.g. ref. 12 and Raven JGM (2019) Crepidula fornicata (Linnaeus, 1758) (Gastropoda: Calyptraeidae) as a hermit crab commensal in the North Sea. Nautilus 133:40–47]

• Line 51 – Desmaulus extinctorium is certainly not restricted to southern China and Hong Kong but has a wide distribution in the Indo-West Pacific.

• Please ensure all references in the text make sense. Either refer to last name, last name et al. or [number]. For example line 55 Regina without any reference number, line 72 Regina et al [17] – referring to the first name of the first author of Cunha et al. [17]. Line 58 Rachel - referring to the first name of Collin [16]. One can only find this by going to the papers referred to. Ditto Line 301 - Alison R. Irwin  Irwin et al. Please check throughout.

• Line 127-135 There is no comment whether any effort weas made to ensure the 51 mitogenomes obtained from GenBank are based on correct identifications. Whilst checking GenBank and underlying studies it is evident the Stromboidea and Naticidae data originate from respectable studies, it would be useful to explicitly state these Littorinimorpha species were analysed by specialists in each group, adding references for each species in Table 1. In many molecular (and other) studies some or a substantial number of the specimens studied have been misidentified. Where authors figure the specimens and list their provenance at least their identity can be verified.

• Line 295-297 – You present this as your conclusion, but for example the species/molecular data and resulting clades for the Stromboidea are virtually the same as used in the previous study [59], only the tree is presented slightly differently. You could be much briefer and indicate that, as expected, your study confirms the conclusions of [59].

• The Naticidae genomes were all taken from Liu et al. (2020) Mitogenomic phylogeny of the Naticidae (Gastropoda: Littorinimorpha) reveals monophyly of the Polinicinae. – but it is not in the references. As is to be extected, the outcome is similar, with some species reversed in the figure. In Liu et al. the key finding (illustrated in their fig. 2) is that the two subfamilies (Naticinae with calcareous operculum and Polinicinae with corneous operculum) are confirmed, but Notocochlis gualtieriana and Notocochlis sp. with calcareous operculum) form a separate clade, making the Naticinae paraphyletic. In your figure 5 Notocochlis is placed within the group Liu et al. consider Polinicinae. If you present it this way you should at least explain why and what the implications are – or not repeat all these data and just select one or two Naticidae.

• Iine 297-298 “At the family level, species of each family formed a distinct clade.” – I assume you mean: Within each superfamily, each family forms a distinct clade.

• The larger number of species from e.g. Naticidae and Littorinidae does not add to understanding placement of Desmaulus within Littorinimorpha. More useful to see data for other Calyptraeidae or potentially closely related families such as Hipponicidae or Capulidae. There is ample genetic material of these in GenBank, but I do not known whether there are any with sufficient mitochondrial DNA to make a good comparison.

• Line 301-302 – These lines confuse what is covered in the study with what genera and species are in each family [this occurs throughout the text]. Assume you want to say: In our study Naticidae is the family of which most species have been included, representing a large number of genera. The Littorinidae are its sister group, of which a substantial number of species has been included in our study, however only representing a selection of the genera.

Minor issues

• Quite a bit of the text suffers from unclear formulations or details that do not really add to the storyline. Some simple editing can improve this. I have selected a few examples.

• Line 12 – For the purpose of augmenting the taxonomy and systematics of Calyptraeidae in the evolutionary framework of Littorinimorpha  why not simply say: For the purpose of determining the placement (or position) of Calyptraeidae within the Littorinimorpha.

• Line 22-25: better say something like: D. extinctorium representing a distinct Calyptraeoidea clade. In summary, this investigation provides the first complete mitochondrial dataset for a species of the Calyptraeidae, thus providing novel insights into the phylogenetic relationships within the Littorinimorpha. (You only analysed one species within this clade, which is formed by many more in 11 genera).

• Line 42: Nassarius members  Nassarius species

• Line 43: Nassariids  nassarids

• Line 46: relationship within gastropod species ??? Do you mean relationship with other gastropod species?

• Line 65: family  order; then also add subclass before Caenogastropoda – or just state that Littorinimorpha comprises 16 superfamilies as the other bit is already stated in line 50.

• Iine 68 - check: do you mean classification OF Littorinimorpha or classification WITHIN Littorinimorpha?

• Line 68: was mainly established by Bouchet [19]. � was established by Bouchet & Rocroi [19].

• Line 79-83: quite unclear. Why not simply state: Riedel [24] established the superfamily Ficoidea, separate from the Tonnoidea, but based on the sequencing of the complete mitochondrial genome of Ficus variegata Wang et al. [23] demonstrated that it fits within the Tonnoidea.

• Line 80 & 81: variegate  variegata [common error caused by the automatic spell checker]

• Line 82 – Riedel reclassified figs – Riedel reclassified fig shells (or even better Ficidae).

• Line 296 – recent families  extant families

6. PLOS authors have the option to publish the peer review history of their article (what does this mean?). If published, this will include your full peer review and any attached files.

Reviewer #1: No

Reviewer #2: **Yes: **Han Raven

---

## [Author Response · Author response to Decision Letter 0]

4 Mar 2024

Dear editor and reviewers,

We thank you for the constructive comments. We have provided detailed responses to all comments below. All revisions to the manuscript have been marked up using the “Track Changes” function in the current version. We are looking forward to your approval or further advice on improving the manuscript, if any.

Major issues 

• A major risk in molecular studies is misidentification of the specimens that were sequenced (there are numerous examples). Therefore it is useful to add photographs of those specimens and indicate where they were collected. As many species included are based on previous or ongoing studies, can you provide the relevant literature references for each? [By the way, I see other papers, e.g. in respectable magazines like Nature, that provide less data than the current study].

Answer: All published references have been cited in Table 1. The genomes of some species have only been uploaded to GenBank for the time being, and the references have not yet been published. As Phylogenetic relationships of the Balkan Moitessieriidae (Caenogastropoda: Truncatelloidea) and Phylogeny of Strombidae (Gastropoda) Based on Mitochondrial Genomes show, some of the genomes involved also have unpublished ones.

• Add at least photographs of the specimen analysed.

Answer: The photographs of the specimen have been added in Fig1.

• Line 49 I have no idea what geological mud is, but suppose you mean siliciclastic mud (instead of carbonate mud) . Anyway, Desmaulus extinctorium is typically found on sandy substrates from low intertidal to several metres subtidally. It is a commensal of hermit crabs, living on the outside of gastropod shells inhabited by these crabs. [own observations and e.g. ref. 12 and Raven JGM (2019) Crepidula fornicata (Linnaeus, 1758) (Gastropoda: Calyptraeidae) as a hermit crab commensal in the North Sea. Nautilus 133:40–47]

Answer: The sentence has been revised.

• Line 51 – Desmaulus extinctorium is certainly not restricted to southern China and Hong Kong but has a wide distribution in the Indo-West Pacific.

Answer: The sentence has been revised.

• Please ensure all references in the text make sense. Either refer to last name, last name et al. or [number]. For example line 55 Regina without any reference number, line 72 Regina et al [17] – referring to the first name of the first author of Cunha et al. [17]. Line 58 Rachel - referring to the first name of Collin [16]. One can only find this by going to the papers referred to. Ditto Line 301 - Alison R. Irwin  Irwin et al. Please check throughout.

Answer: These details have been reviewed and modified.

• Line 127-135 There is no comment whether any effort weas made to ensure the 51 mitogenomes obtained from GenBank are based on correct identifications. Whilst checking GenBank and underlying studies it is evident the Stromboidea and Naticidae data originate from respectable studies, it would be useful to explicitly state these Littorinimorpha species were analysed by specialists in each group, adding references for each species in Table 1. In many molecular (and other) studies some or a substantial number of the specimens studied have been misidentified. Where authors figure the specimens and list their provenance at least their identity can be verified.

Answer: All published references have been cited in Table 1. The genomes of some species have only been uploaded to GenBank for the time being, and the references have not yet been published. As Phylogenetic relationships of the Balkan Moitessieriidae (Caenogastropoda: Truncatelloidea) and Phylogeny of Strombidae (Gastropoda) Based on Mitochondrial Genomes show, some of the genomes involved also have unpublished ones.

• Line 295-297 – You present this as your conclusion, but for example the species/molecular data and resulting clades for the Stromboidea are virtually the same as used in the previous study [59], only the tree is presented slightly differently. You could be much briefer and indicate that, as expected, your study confirms the conclusions of [59].

Answer: This part has been revised.

• The Naticidae genomes were all taken from Liu et al. (2020) Mitogenomic phylogeny of the Naticidae (Gastropoda: Littorinimorpha) reveals monophyly of the Polinicinae. – but it is not in the references. As is to be extected, the outcome is similar, with some species reversed in the figure. In Liu et al. the key finding (illustrated in their fig. 2) is that the two subfamilies (Naticinae with calcareous operculum and Polinicinae with corneous operculum) are confirmed, but Notocochlis gualtieriana and Notocochlis sp. with calcareous operculum) form a separate clade, making the Naticinae paraphyletic. In your figure 5 Notocochlis is placed within the group Liu et al. consider Polinicinae. If you present it this way you should at least explain why and what the implications are – or not repeat all these data and just select one or two Naticidae.

Answer: In our figure 5, Notocochlis is not placed within Polininae; instead, it remains a species within Naticinae. The divergence arises because our study selected a broader range of Naticidae species genomes from NCBI to construct the phylogenetic tree. Consequently, the branch of Naticinae is divided into two parts, indicating that Naticinae lacks monophyly. In Liu et al.'s study, there is no explicit confirmation of the monophyly of Naticinae either. Subsequent investigations may require additional genomic data from more Naticidae species to address this issue. Therefore, these details are not elaborated upon in the manuscript.

• Iine 297-298 “At the family level, species of each family formed a distinct clade.” – I assume you mean: Within each superfamily, each family forms a distinct clade.

Answer: The sentence has been revised.

• The larger number of species from e.g. Naticidae and Littorinidae does not add to understanding placement of Desmaulus within Littorinimorpha. More useful to see data for other Calyptraeidae or potentially closely related families such as Hipponicidae or Capulidae. There is ample genetic material of these in GenBank, but I do not known whether there are any with sufficient mitochondrial DNA to make a good comparison.

Answer: Due to the exclusive utilization of complete mitochondrial genomes in this study for constructing the phylogenetic tree, there is currently an absence of complete mitochondrial genome information for species within Hipponicidae in the GenBank database. For Capulidae species, there is only one complete mitochondrial genome available, uploaded in February 2024, which is insufficient for a comprehensive comparative analysis. As a result, the study does not include a phylogenetic analysis of Hipponicidae and Capulidae.

• Line 301-302 – These lines confuse what is covered in the study with what genera and species are in each family [this occurs throughout the text]. Assume you want to say: In our study Naticidae is the family of which most species have been included, representing a large number of genera. The Littorinidae are its sister group, of which a substantial number of species has been included in our study, however only representing a selection of the genera.

Answer: This part has been revised.

Minor issues

• Quite a bit of the text suffers from unclear formulations or details that do not really add to the storyline. Some simple editing can improve this. I have selected a few examples.

Answer: The sentence has been revised.

• Line 12 – For the purpose of augmenting the taxonomy and systematics of Calyptraeidae in the evolutionary framework of Littorinimorpha  why not simply say: For the purpose of determining the placement (or position) of Calyptraeidae within the Littorinimorpha.

Answer: The sentence has been revised.

• Line 22-25: better say something like: D. extinctorium representing a distinct Calyptraeoidea clade. In summary, this investigation provides the first complete mitochondrial dataset for a species of the Calyptraeidae, thus providing novel insights into the phylogenetic relationships within the Littorinimorpha. (You only analysed one species within this clade, which is formed by many more in 11 genera).

Answer: The sentence has been revised.

• Line 42: Nassarius members  Nassarius species

Answer: The word has been revised.

• Line 43: Nassariids  nassarids

Answer: The word has been revised.

• Line 46: relationship within gastropod species ??? Do you mean relationship with other gastropod species?

Answer: The sentence has been revised.

• Line 65: family  order; then also add subclass before Caenogastropoda – or just state that Littorinimorpha comprises 16 superfamilies as the other bit is already stated in line 50.

Answer: The sentence has been revised.

• Iine 68 - check: do you mean classification OF Littorinimorpha or classification WITHIN Littorinimorpha?

Answer: The sentence has been revised.

• Line 68: was mainly established by Bouchet [19]. � was established by Bouchet & Rocroi [19].

Answer: The sentence has been revised.

• Line 79-83: quite unclear. Why not simply state: Riedel [24] established the superfamily Ficoidea, separate from the Tonnoidea, but based on the sequencing of the complete mitochondrial genome of Ficus variegata Wang et al. [23] demonstrated that it fits within the Tonnoidea.

Answer: The sentence has been revised.

• Line 80 & 81: variegate  variegata [common error caused by the automatic spell checker]

Answer: The sentence has been revised.

• Line 82 – Riedel reclassified figs – Riedel reclassified fig shells (or even better Ficidae).

Answer: The sentence has been revised.

• Line 296 – recent families  extant families

Answer: The word has been revised.

---

## [Decision Letter · Decision Letter 1]

14 Mar 2024

Characterization of the complete mitochondrial genome of Desmaulus extinctorium (Littorinimorpha, Calyptraeoidea, Calyptraeidae) and molecular phylogeny of Littorinimorpha

PONE-D-23-34396R1

Dear Dr. Ye,

We’re pleased to inform you that your manuscript has been judged scientifically suitable for publication and will be formally accepted for publication once it meets all outstanding technical requirements.

Kind regards,

Sean Michael Prager, Ph.D.

Academic Editor

PLOS ONE

Additional Editor Comments (optional):

Reviewers' comments:

Reviewer's Responses to Questions

**Comments to the Author**

1. If the authors have adequately addressed your comments raised in a previous round of review and you feel that this manuscript is now acceptable for publication, you may indicate that here to bypass the “Comments to the Author” section, enter your conflict of interest statement in the “Confidential to Editor” section, and submit your "Accept" recommendation.

Reviewer #2: All comments have been addressed

2. Is the manuscript technically sound, and do the data support the conclusions?

Reviewer #2: Yes

3. Has the statistical analysis been performed appropriately and rigorously? 

Reviewer #2: Yes

4. Have the authors made all data underlying the findings in their manuscript fully available?

Reviewer #2: Yes

5. Is the manuscript presented in an intelligible fashion and written in standard English?

Reviewer #2: Yes

6. Review Comments to the Author

Reviewer #2: Excellent follow up on my comments. I just found some really minor items:

Line 51. D. extinctorium �– At the start of a sentence always write the full name of the genus.

Line 52. is primarily distributed in the southern � is abundant in southern

Line 76 typo in the word additionally

Please go ahead and publish.

7. PLOS authors have the option to publish the peer review history of their article (what does this mean?). If published, this will include your full peer review and any attached files.

Reviewer #2: **Yes: **Han Raven

---

## [Editor Report · Acceptance letter]

20 Mar 2024

PONE-D-23-34396R1 

PLOS ONE

Dear Dr. Ye, 

I'm pleased to inform you that your manuscript has been deemed suitable for publication in PLOS ONE. Congratulations! Your manuscript is now being handed over to our production team.

Kind regards, 

on behalf of

Dr. Sean Michael Prager 

Academic Editor

PLOS ONE